# Examination of 2D frontal and sagittal markerless motion capture: Implications for markerless applications

**Logan Wade** *, Laurie Needham, Murray Evans, Polly McGuigan, Steffi Colyer, Darren Cosker, James Bilzon

Centre for the Analysis of Motion, Entertainment Research and Applications, University of Bath, Bath, United Kingdom

* lw2175@bath.ac.uk

## Abstract

This study examined if occluded joint locations, obtained from 2D markerless motion capture (single camera view), produced 2D joint angles with reduced agreement compared to visible joints, and if 2D frontal plane joint angles were usable for practical applications. Fifteen healthy participants performed over-ground walking whilst recorded by fifteen marker-based cameras and two machine vision cameras (frontal and sagittal plane). Repeated measures Bland-Altman analysis illustrated that markerless standard deviation of bias and limits of agreement for the occluded-side hip and knee joint angles in the sagittal plane were double that of the camera-side (visible) hip and knee. Camera-side sagittal plane knee and hip angles were near or within marker-based error values previously observed. While frontal plane limits of agreement accounted for 35–46% of total range of motion at the hip and knee, Bland-Altman bias and limits of agreement (-4.6–1.6 ± 3.7–4.2˚) were actually similar to previously reported marker-based error values. This was not true for the ankle, where the limits of agreement (± 12˚) were still too high for practical applications. Our results add to previous literature, highlighting shortcomings of current pose estimation algorithms and labelled datasets. As such, this paper finishes by reviewing methods for creating anatomically accurate markerless training data using marker-based motion capture data.

## Introduction

The agreement between markerless motion capture and marker-based motion capture for biomechanical applications has been steadily improving over the past two decades [1–3]. There have been several highly detailed reviews of current markerless motion capture methods [1–3], detailing the different approaches used, and highlighting the systematic differences observed between markerless motion capture and manual-labelled or marker-based motion capture. Markerless motion capture has the potential to streamline data collection, enabling capture of human movement with minimal burden placed on the participant [3], although the costs of current commercial systems are still highly prohibitive for developing countries and smaller

accross 14 participants), in both the sagittal and frontal plane, for the markerless system and the reprojected marker-based motion capture system. The JSON file format is universally implemented.

**Funding:** This research was funded by the EPSRC, through CAMERA, the RCUK Centre for the Analysis of Motion, Entertainment Research and Applications (EP/ M023281/1 and EP/T022523/1).

**Competing interests:** The authors have declared that no competing interests exist.

clinical applications. Open-source markerless motion capture, paired with low-cost cameras, provides researchers, clinicians and coaches with an accessible, cost-effective and transparent motion capture solution [4–9], enabling analysis of clinical and sporting gait parameters without the need for specialist personnel to process the data [5, 10]. Furthermore, future pose estimation algorithms could be employed to extract additional key variables from large clinical video databases that have been acquired over many years [5, 10]. Previous 2D markerless motion capture research has examined sagittal plane kinematics during overground and treadmill walking, and underwater running [4, 6, 11, 12], identifying joint angles errors that are in line with those obtained when using manually labelled or marker-based motion capture methods. However, to date, only angles of joints nearest to the camera have been examined [11, 12], despite pose estimation algorithms often providing joint centre locations on the occluded far side of the body [13]. Current open access pose estimation training datasets have been labelled with both visible key point locations (i.e. joint centre locations that can directly be seen from a 2D camera view), and occluded key point locations (i.e. joint centre locations that are obstructed by another body part from a 2D camera view) [14, 15], which is useful for entertainment applications, ensuring that detected body parts do not simply disappear when occluded by other limbs. However, from a biomechanical point of view, manually labelling occluded joints requires the labeller to estimate where the joint centre is located, likely resulting in increased errors and uncertainty. While occluded joints may have greater uncertainty compared to visible joints, the difference could potentially be small enough to have a minimal effect for some applications, especially in only partially occluded joints (e.g., ankle and knee during gait). To date, there has been no comparison of the agreement between joint angles obtained from occluded and visible joints, when using 2D open-source pose estimation algorithms and marker-based motion capture.

Frontal plane joint angles and subsequently, frontal plane joint kinetics, provide valuable insights into abnormal movement and loading for many sporting and clinical applications, with frontal plane mechanics of the lower limb a particular interest for disorders such as knee osteoarthritis [16, 17] and stroke [18], as well as in sporting movements such as jumping and landing [19, 20]. In the 2D frontal plane, joint occlusion is minimal, however, limited movement requires high accuracy and low variability to detect small but meaningful changes. To our knowledge, only one study has examined markerless frontal plane joint angles, examining 2D frontal plane hip angle during treadmill walking with the OpenPose pose estimation algorithm, finding relatively large differences compared to a 3D marker-based system [12]. However this study defined an unusual hip angle (angle between opposite hip-hip-knee) and did not examine knee or ankle frontal plane angles [12]. As such, there has currently been very limited evaluation of frontal plane joint angles using 2D markerless methods, leading to uncertainty of usability for clinical and sporting applications.

This paper aims to explore two interconnected questions by comparing 2D markerless motion capture, which uses the OpenPose single camera 2D pose estimation (OpenPose), against marker-based motion capture. Firstly, are markerless occluded joint angles and joint centre locations identified with worse agreement than visible joints, relative to marker-based motion capture methods? Secondly, do markerless frontal plane joint angles have sufficient agreement with marker-based motion capture for practical applications? It was hypothesised that occluded joint angles in the sagittal view would have greater bias and limits of agreement compared visible joints, relative to marker-based motion capture. It was also hypothesised that markerless motion capture will likely have limits of agreement that are too high compared to marker-based motion capture to detect meaningful changes in frontal plane joint angles. Finally, this paper will finish by reviewing methods for creating anatomically accurate markerless training data (labelled images) using marker-based motion capture data.

## Methods

Fifteen healthy participants (7 males and 8 females, 26 ± 5 years old, 173 ± 11 cm, 73 ± 14 kg) gave written informed consent to perform overground preferred constant speed walking, while markerless and marker-based motion capture data were recorded. Ethics were approved by the Research Ethics Approval Committee for Health at the University of Bath (EP1819052), and all methods were performed in accordance with the University of Bath guidelines and regulations. Written informed consent was obtained from all participants regarding the publication of open-access identifiable images and videos. As such, the individual pictured in Fig 1 has provided written informed consent to publish their image alongside the manuscript. Data collection was performed as part of a larger study that recorded synchronised 3D marker-based motion capture and 3D markerless motion capture, which is described in depth by Needham, Evans [21]. For this this study, motion capture data (200 Hz) from fifteen Qualisys cameras (Oqus, Qualysis, Gothenburg, Sweden) and two machine vision cameras (1920x1080 pixels, JAI sp5000c, JAI ltd, Denmark) were analysed. As can be seen in Fig 1, one machine vision camera was set up in the sagittal plane (right hand side of the body during walking) and one was set up in the frontal plane (directly in front of the participant). Video data from both systems were time synchronised using transistor-transistor logic (TTL) pulses sent out from the markerless system. To ensure that there was no drift in the synchronisation, two visible

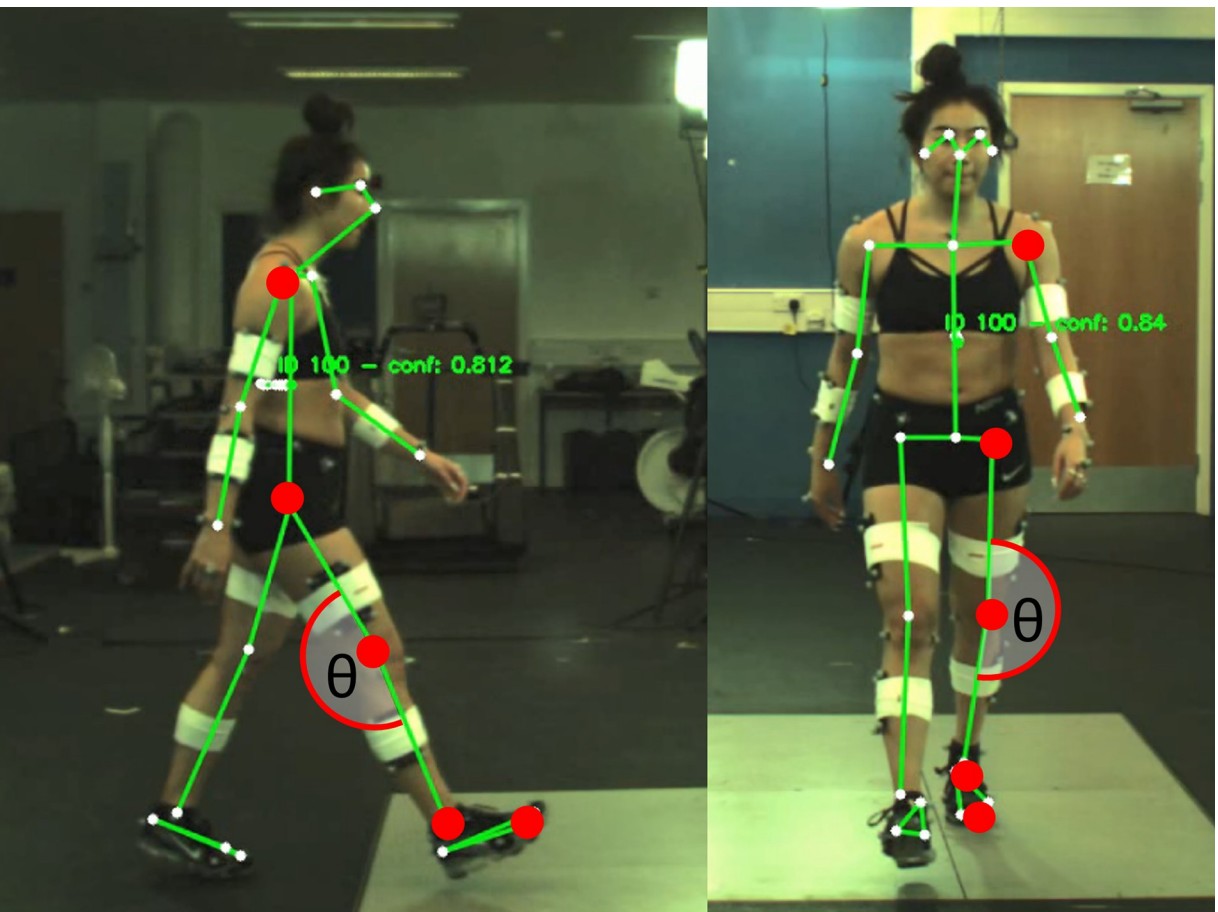

**Fig 1. Sagittal and frontal plane camera views, with example joint centre locations of the metatarsophalangeal, ankle, knee, hip and shoulder, and how they are used to calculate joint angle of the knee in the frontal and sagittal plane.**

LED lights and one infra-red LED were placed in the capture volume of both camera system views, which flashed and ensured that frame alignment between systems was correct [21]. Calibration of Qualisys cameras were performed as per the manufacturer specifications. Calibration of the markerless camera system used observations of a binary dot matrix to identify each cameras intrinsic parameters, then extrinsic parameters were identified from pairs of cameras with shared dot matrix observations [21]. Euclidean spaces of both systems were aligned by moving a single marker through both camera spaces to match corresponding marker locations [21]. 3D marker-based motion capture data was collected using 44 individual retroreflective markers and eight retroreflective marker clusters attached to the participants' upper and lower limbs, pelvis and torso, described in detail by Needham et al. [21].

2D markerless motion capture video data for the sagittal and frontal camera views were processed using the OpenPose open-source pose estimation algorithm [13], which identified 25 keypoints on the participant (Fig 1). While OpenPose is capable of multi-person detection in a single image, it does not associate or 'track' people between frames, therefore the person ID given to each individual within an image may change frame to frame. To track only our participant and ignore additional people or false detections, we make use of a classic and highly efficient Kalman filter and the Hungarian algorithm approach [22, 23]. Each detected person's keypoints are reduced to a single median point and represented by a linear, constant-velocity motion model and confidence value. The confidence threshold was set to 60%, below this value often represented false positive detections, caused by tripods in the background. The motion model, based on a Kalman filter, is then used to propagate a person's identity in the next frame. The state of the model can be represented as

$$x = [u, v, s, u', v', s']^T$$

Where u and v represent 2D pixel coordinate locations on the image plane, s is the scale of the detected persons and u', v' and s' represent the rate of change of these respective values between frames. When the detection is associated to a currently tracked person, points u and v are used to update the target state where the velocity components are solved optimally within the Kalman filter. To assign detections to existing persons, each person's median point is estimated by predicting the new location in the current frame. The assignment cost matrix is then computed as the Euclidean distance between each detection and all the predicted locations of the existing persons. The assignments are then optimally solved using the Hungarian algorithm [24]. Once all frames had been processed, the participant's tracked data were collated within a single ID that could be identified from the list of tracked persons, based on the tracked ID with the greatest cumulative distance in pixels, due to the participant being the only person moving substantially during a trial.

Markerless 2D sagittal and frontal plane joint centre locations over the entire trial were smoothed using a bi-directional Kalman filter, which accounts for previous and future locations of each marker. This filtering method has previously been shown to be more effective on markerless data than traditional biomechanical low pass filtering [21]. Markerless shoulder, hip, knee and ankle joint centre locations were obtained directly from the pose estimation algorithm output, while the metatarsophalangeal (MTP) joint was defined as the midpoint between the first and fifth toe keypoints. Marker-based data were collected and processed in Qualisys Track Manager (Qualisys, Gothenburg, Sweden) before being exported to Visual3D (C-Motion Inc, Germantown, USA). Hip joint centres were computed via regression from the anterior and posterior superior iliac spine markers [25], knee and ankle joint centres were computed as the midpoint between the medial and lateral markers, shoulder joint centres were

inferiorly offset from the acromioclavicular joint shoulder markers by 2.5 cm, and the MTP joint centres were computed as the midpoint between the first and fifth MTP joint markers.

Because markerless and marker-based cameras generally cannot occupy the same physical space, previous research has compared results between systems using two separate cameras placed next to each other, resulting in only an approximate alignment between the planes of each system's Euclidean spaces [26, 27]. This will likely result in some parallax error between cameras that may manifest as systematic differences between the two systems. To mitigate this issue and provide an aligned comparison between markerless and marker-based results, 3D marker-based joint centre locations were reprojected within each 2D markerless cameras' Euclidean space (sagittal and frontal camera views) [28]. Reprojected marker-based joint centre coordinates were then filtered using a bi-directional Butterworth low-pass filter (10 Hz). Once markerless and marker-based joint centre locations were filtered, event timings (heel-strike and toe-off) were obtained from Visual 3D using ground reaction force data from four imbedded force plates (Kistler 9287CA, Winterthur, Switzerland) to ensure that gait events were identical for both marker-based and markerless systems. Markerless and reprojected marker-based joint centre locations were trimmed and time normalised (101 points) to one stride (heel-strike to heel-strike) for the right and left legs separately. Left and right, ankle, knee and hip planar joint angles for both marker-based and markerless methods were calculated using the MTP, ankle, knee, hip and shoulder joint centre locations, for the frontal and sagittal views (Fig 1). Because the participant is walking towards the frontal camera, their scale is changing in every frame over one stride, which may influence pixel location differences between the 2D markerless motion capture and the reprojected marker-based motion capture methods. To account for this, all pixel differences were normalised to the average vertical distance in pixels (height) between the markerless left and right, shoulder and hip joint centre locations of the participant, for every frame, in each individual trial. Averaged left and right, hip and shoulder distances were used as these joint locations remain in the same plane as the centre of mass for both frontal and sagittal plane movements. It should be noted that while this method will minimise errors as the participants scale changes by getting closer to the camera, it does not account for the left and right limbs being closer or further away from the camera at different points in the stride. However, because we asked participants to hit the first force plate with their right foot five times, and then ask them to switch feet by hitting the first force plate with their left foot five times, this should not affect one side of the body more than the other. While this normalisation was not likely needed for the sagittal plane, it will help to minimise any parallax error and provides comparative values between planes, therefore it was performed for both frontal and sagittal plane joint centre location results.

Statistical analysis was performed on the left and right; ankle, knee and hip angles, and MTP, ankle, knee, hip and shoulder joint centre locations in the frontal and sagittal plane, with reprojected marker-based results as the reference method [29]. Insufficient bias, standard deviation (SD) of bias and limits of agreement of the markerless system, relative to the marker-based system were defined as differences that were greater than known marker-based errors previously reported [30–32]. To clearly describe differences in the sagittal plane, joint angles and locations in this plane will henceforth be referred to as camera-side (right side joints) and occluded-side (left side joints). Due to the repeated measures performed with each participant (up to 10 trials, each with 101 data points), repeated measures Bland-Altman analysis where the true value varies (i.e. joint angles changes over one stride), was employed to ensure that SD's were not underestimated [33]. For each joint angle and joint centre location outcome variable, this method calculates the SD, and subsequently the upper and lower limits of agreement (95% confidence interval), using total variance across all data points from a one-way ANOVA (participant and residual mean square scores). In-depth detail and rationale of the repeated

measures Bland-Altman analysis, with equations and examples, can be found in S1 Appendix and S1 Data. Repeated measures correlation was also performed on joint angle values to examine the correlation coefficient between markerless and marker-based methods. This analysis was performed using the python implementation of the RStudio rmcorr package [34, 35]. Each participant completed 10 preferred speed walking trials for a total of 150 trials (15 participants x 10 trials). However due to marker-drop out, or incorrect foot placement (i.e., the entire foot was not placed on the force plate), 130 successful trials were analysed. The average preferred walking speed across all participants was 1.55 ± 0.23 m/s. Percentage differences reported below are the percent difference of the markerless bias relative to the marker-based motion capture joint range of motion for individual joints.

## Results

### Sagittal plane joint centre location

On average, systematic differences for the camera-side and occluded-side joint centre locations of the ankle and knee were similar (Fig 2A, S1 Table), while occluded-side hip and shoulder joint centre locations (Fig 2A, S1 Table) were worse than their camera-side counterparts (Fig 2A, S1 Table). For both sides of the body, repeat measures Bland-Altman bias and SD of bias in the ankle and knee had substantial spikes during push-off and swing phase (Fig 2C and 2D), while the hip and shoulder joint centre locations were relatively constant over the stride (Fig 2A, 2E and 2F). Bias and SD of bias of the MTP joint for both the camera (Fig 2A, S1 Table) and occluded-side (Fig 2A, S1 Table) were worse than all other joints (Fig 2A), which was produced primarily during the push-off and swing phase of walking (Fig 2B). Sagittal plane repeated measures Bland-Altman results, for joint centre locations, can be found in S1 Table.

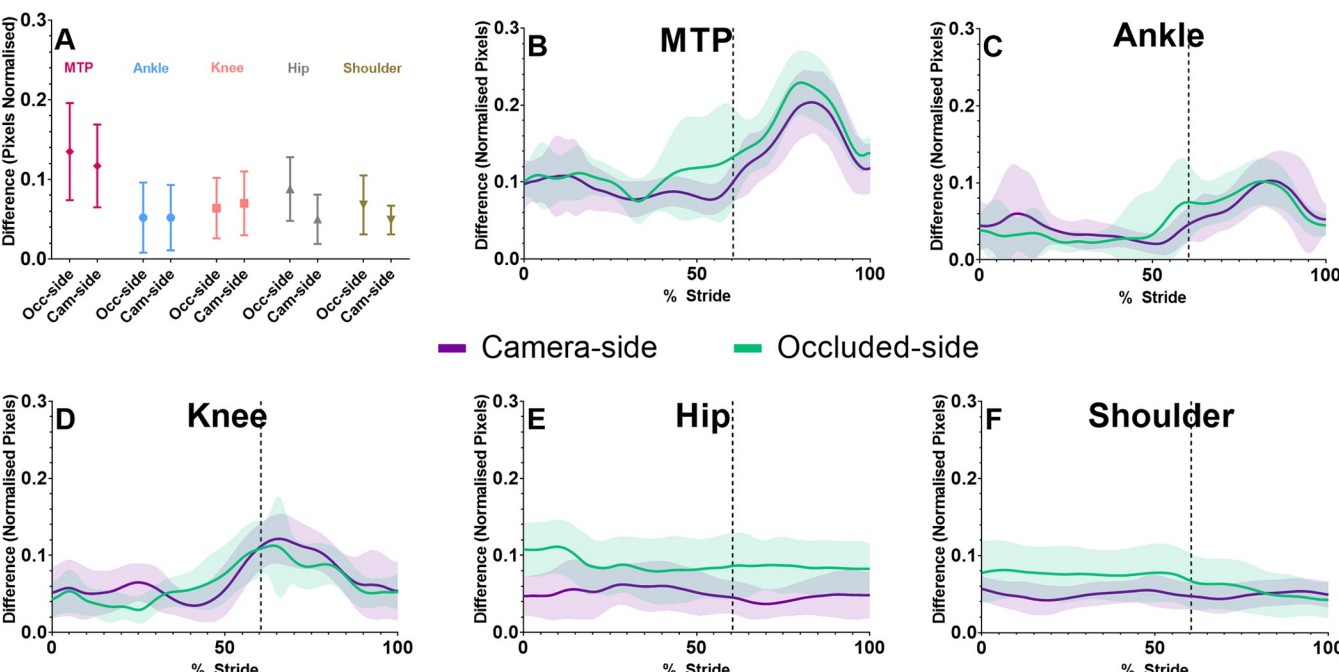

**Fig 2.** Sagittal plane joint centre location differences over the whole stride (A) for the camera-side (Cam-side) and occluded-side (Occ-side), created using repeated measures Bland-Altman bias and SD of bias,. In Fig 2B–2F, frontal plane joint centre locations were time normalised to one stride and then, within stride mean and SD differences were calculated between markerless and marker-based motion capture for each joint, averaged over all trials. Vertical lines in Fig 2B-2F represents the end of the stance phase (60.4% of stride). All values were normalised to the vertical distance in pixels (height) between the left and right shoulder and hip joint centre locations of the participant, for every frame in each individual trial.

**Table 1. Repeated measures Bland-Altman analysis and repeated measures correlation coefficient of the 2D markerless ankle, knee and hip joint angles in the sagittal and frontal plane, relative to reprojected marker-based motion capture.** Camera-side represents the right side of the body, which was closest to the camera, while the occluded-side represents the left side of the body, which was furthest from the camera in the sagittal view. Left and right sides of the body were combined in the frontal plane.

| Joint Angles | | Bias (°) | SD of Bias (°) | Limits of Agreement (°) | Correlation Co-efficient (r2) |
|---|---|---|---|---|---|
| **Sagittal Plane** | | | | | |
| Ankle | Camera-side | -8.4 | 5.2 | -18.6–1.9 | 0.82 |
| | Occluded-side | -9.3 | 7.3 | -23.6–5.1 | 0.65 |
| Knee | Camera-side | 1.5 | 4.1 | -6.5–9.6 | 0.98 |
| | Occluded-side | 1.6 | 6.9 | -12.0–15.2 | 0.94 |
| Hip | Camera-side | -3.6 | 4.6 | -12.6–5.3 | 0.94 |
| | Occluded-side | -4.6 | 9.5 | -23.2–14.0 | 0.76 |
| **Frontal Plane** | | | | | |
| Ankle | | 0.2 | 12.0 | -23.4–23.8 | 0.26 |
| Knee | | 1.6 | 4.2 | -6.7–9.9 | 0.73 |
| Hip | | -4.6 | 3.7 | -11.5–3.0 | 0.46 |

## Sagittal plane joint angle

Sagittal plane systematic differences (bias) across the ankle, knee and hip were relatively consistent between camera and occluded sides (slightly higher for the occluded side). However, the Bland-Altman SD of bias were 40%, 69% and 108% greater in the occluded ankle, knee and hip respectively, compared to their camera-side counterparts (Table 1, Fig 3A–3C). Average joint angle range of motion (RoM) obtained from the marker-based method was 37 ± 6 °,

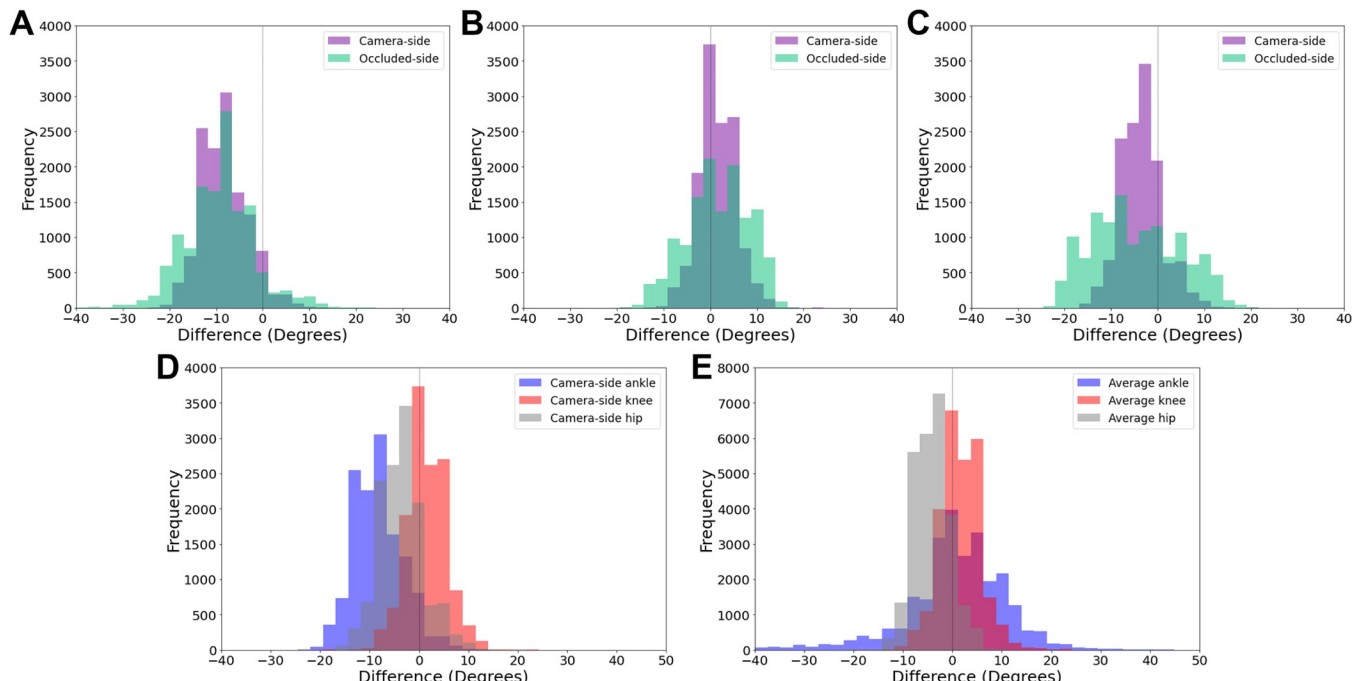

**Fig 3. Distribution of joint angle differences between markerless and marker-based methods, for all individual data points (101 timepoints x successful trials x participants).** Fig 3A–3C illustrate the differences between camera-side and occluded side joints of the ankle (A), knee (B) and hip (C). D illustrates the camera-side only differences for the ankle, knee and hip. Fig 3E illustrates the differences for the combined left and right ankle, knee and hip in the frontal plane.

$45 \pm 6°$ and $67 \pm 4°$ for the ankle, knee and hip respectively. As such, markerless SD of bias accounted for 14% and 20% of total ankle joint RoM on the camera-side and occluded-side joint angles respectively. Alternatively, in the knee and hip, markerless SD of bias accounted for 7–9% RoM in camera-side joints and 14–15% RoM in occluded-side joints. Timeseries data of sagittal plane joint angles and differences are presented in S1 Fig.

### Frontal plane joint centre location

Systematic differences of 2D joint centre locations were relatively consistent across all joints, being lowest at the ankle and highest at the shoulder (Fig 4A, S1 Table). However, SD of bias of joint centre locations was greatest at the MTP joint (Fig 4A, S1 Table) and decreased as joint level increased, being smallest at the shoulder (Fig 4A, S1 Table). Examination of bias and SD of bias relative to marker-based motion capture indicated that there was a large increase after push-off in the MTP, ankle and knee joint centre locations (Fig 4B), while the hip and shoulder were relatively similar throughout the stride. Frontal plane repeated measures Bland-Altman table for joint centre locations can be found in S1 Table.

### Frontal plane joint angle

Bias and SD of bias were relatively similar in the frontal plane compared with the sagittal plane about the knee and hip (Table 1). However, while the bias was improved about the ankle angle relative to the sagittal plane, the SD of bias was substantially increased and the correlation coefficient was substantially reduced (Table 1, Fig 3E). Mean total joint RoM from marker-based data was $31 \pm 15°$ for the ankle, $12 \pm 5°$ for the knee and $8 \pm 3°$ for the hip. Due to the small RoM at the hip and the knee, and the large SD of bias at the ankle, frontal plane markerless SD of bias accounted for 39%, 35% and 46% of the total RoM about the ankle, knee and hip respectively. Timeseries data of frontal plane joint angles and differences are presented in S1 Fig.

### Discussion

Compared to camera-side joints, SD of bias and limits of agreement for the sagittal plane occluded joint angles were higher in more proximal joints, with the hip being highest (Table 1,

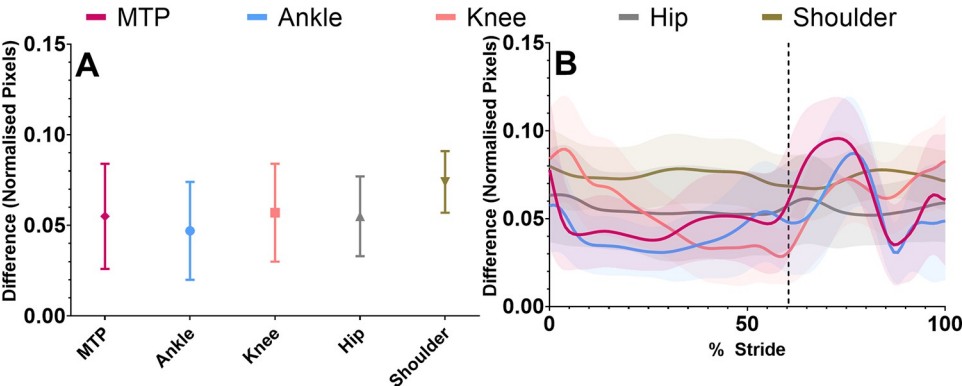

**Fig 4.** Frontal plane joint centre locations over the whole stride (A), calculated using repeated measures Bland-Altman bias and SD of bias. Frontal plane joint centre locations were time normalised and within stride mean and SD differences were calculated (B) between markerless and marker-based motion capture for each joint averaged over all trials. Vertical lines in Fig 4B represent end of the stance phase (toe-off). All values were normalised to the vertical distance in pixels (height) between the left and right shoulder and hip joint centre locations of the participant, for every frame in each individual trial.

Fig 3). Because occluded-side hip and shoulder joint centre locations were obstructed for the entire stride, they were more affected than the MTP, ankle and knee (Fig 2). Therefore, the occluded-side knee angle had lower limits of agreement, as it was calculated using vector angles between the ankle, knee and hip joint centre locations, compared to the hip angle, which used the knee, hip and shoulder joint centre locations. The knee and hip ratio of SD of bias, to total joint RoM, was relatively small in the camera-side joints (7–9%), while the occluded-side knee and hip joints were almost double (15%). Alternatively at the ankle, the SD of bias was only slightly higher in the occluded-side compared to the camera side (Table 1). Overall, the ankle ratio of SD of bias, to total joint RoM, was higher than the knee and hip, which was likely due to two causes. Firstly, the RoM about the ankle is much lower than the knee and hip, and thus, the relative proportion of error to signal is higher. Secondly, the markerless system identifies the end of the foot by labelling the hallux (first digit) and fifth digit of the foot, while marker-based motion capture uses the first and fifth MTP joint. Therefore, the markerless ankle joint angle will be influenced by toe extension during push-off [36]. These results indicate that the occluded hip and knee joints have roughly double the variability of the camera-side joints, and overall, the ankle angle has considerable bias due to biomechanically limited labelling of the foot in the COCO open access training dataset [15] that is used by the OpenPose algorithm [13] we employed.

Frontal plane joint angle bias were relatively small for the ankle and knee, while the hip exhibited a larger offset (Fig 3), although this is most likely due to the differences in systematic bias previously identified in this markerless training dataset [21], due to systematic differences of joint centre locations identified by human labellers and marker-based motion capture [21]. The ankle angle had considerably worse SD of bias, limits of agreement and correlation coefficient than the hip and knee (Fig 3E and Table 1), which was due to the MTP, ankle and knee joint centre locations having increased differences during push-off, swing phase and heel-strike (Fig 4B), when knee flexion, dorsiflexion and toe extension may produce occlusion of the MTP and ankle joint. Previous research exploring out of sagittal plane movements from a 2D sagittal plane camera had some success by combining markerless methods with IMU devices, which could be a short term solution to improving the results at the ankle using current markerless motion capture datasets [37]. While we did observe a systematic offset (bias) in the hip frontal angle between the markerless and marker-based methods (S1 Fig), this difference was constant throughout the gait cycle. Thus, hip frontal angle bias relative to marker-based motion capture was lower than the results of Ota et al., [12], who found much larger hip frontal angle differences, especially during swing. This was most likely caused the unusual frontal hip joint angle definition performed by Ota et al., [12], as they defined the frontal hip angle using vector angles between the contralateral hip, ipsilateral hip and ipsilateral knee joint centre locations [12]. We defined the hip angle using vector angles between the ipsilateral shoulder, hip and knee joint centre locations. Thus, the stance and swing of the opposite leg may have affected Ota et al's frontal plane hip angle, which would not affect our results as we used joint centre locations from only one side of the body (Fig 1).

These results highlight several implications of open-source 2D markerless motion capture (OpenPose) for clinical and sporting applications. Firstly, SD of bias and limits of agreement were almost double in the occluded-joint angles compared to visible camera-side joints, and therefore, occluded joints should be discounted when practitioners and coaches are assessing results. Although it should be noted that the correlation coefficient of the occluded knee was only slightly worse than the intact knee and the occluded knee it was much better than the occluded hip, thus the occluded knee joint angle may not be as susceptible to occlusion issues as the hip. Secondly, both occluded and camera-side MTP, ankle and knee joint centres had increased bias relative to marker-based motion capture during the swing phase, thus, these

periods may currently need to be omitted to facilitate examination of joint angles. Thirdly, the MTP joint definition from OpenPose does not account for toe flexion during push-off, which further reduces the agreement of the ankle joint angle [36]. Finally, in 2006, Benoit et al. [32] used bone-pin markers to examine the error in knee joint angle due to skin marker movement during walking, finding marker-based knee flexion/extension errors of 2.8 ± 2.6˚ and knee abduction/adduction errors of 4.4 ± 3.2˚. In comparison, relative to marker-based motion capture, our markerless knee angle difference in the sagittal plane was 1.5 ± 4.1˚ (flexion/extension), and in the frontal plane was 1.6 ± 4.2˚ (adduction/abduction), thus markerless differences are likely near or within marker-based error rates for the knee in the sagittal (camera-side only) and frontal plane. It should be noted that Benoit, Ramsey [32] obtained absolute error values at foot-strike, mid-stance and toe-off, while our study examined the entire stride, finding higher SD of joint centre locations during the swing phase (Fig 2). Therefore, the differences we observed may likely reduce further if we only examined the stance phase as was done by Benoit et al. [32]. Due to similar RoM and differences for the knee and hip angle in our study, these justifications likely extend to the hip as well. Unfortunately, markerless ankle angle differences compared to marker-based motion capture were still too high when compared to a study that examined ankle marker-based errors against biplanar fluoroscopy [31]. This biplanar study observed ankle plantarflexion errors ranging maximally up to 4.2˚ and frontal plane maximal errors ranging up to 6.3˚, while our study observed markerless 95% limits of agreement ranging up to 18.6˚ in the sagittal plane and 23.8˚ in the frontal plane. Due to these known marker-based errors, comparisons between marker-based motion capture and markerless motion capture only give us a strong estimate of the accuracy and variability of markerless motion capture. Our findings suggest that for clinical, sporting and research applications during walking, 2D sagittal and frontal plane kinematics identified by the OpenPose algorithm could be obtained for the visible knee and hip joint angles during stance. Alternatively, the sagittal ankle angle is likely only usable during early and mid-stance, and the frontal plane ankle angle differences were too high across the whole stride. To determine the true accuracy and variability of markerless systems, comparisons against gold standard measures such as bone pins and biplanar fluoroscopy may be required, although differing coordinate system definitions may still exist between systems that could result in systematic differences.

The results from this study also have implications for 3D markerless motion capture systems that are trained on open access dataset, due to observed difference between occluded and visible (camera-side) joint centre locations. Manual labelling within pose estimation training datasets requires human labellers to estimate the location of occluded joints, thus, it is not surprising that occluded joints identified using pose estimation algorithms will also have increased uncertainty. The inclusion of occluded joints will therefore increase the bias and limits of agreement of the 3D fusion solution from multiple 2D camera views, especially compared to a system that only performs fusion using joint centre locations of visible joints. Unfortunately, open-source pose estimation algorithms such as OpenPose [13] generally do not appear to differentiate between visible and occluded joints, despite some datasets providing this information [15, 38]. Additionally, Cronin [39] suggested that the inclusion of indiscriminate occluded joints when training the pose estimation algorithm might also reduce the accuracy of detecting all joints, due to the pose estimation algorithm learning that a joint centre location may appear on a point of the body that is not the actual joint. Assessing this theory requires comparison of two identical pose estimation algorithms that differ only in their training dataset (i.e., trained with and without occluded joints), which was beyond the scope of this paper. As such, creation of anatomically accurate markerless datasets should seek to label and classify both occluded and visible keypoints [15, 38], which may facilitate creation of highly accurate training datasets while also making these datasets functional for broader applications (e.g. virtual reality).

Our research supports previous papers [3, 21, 36, 39, 40] highlighting the need of anatomically accurate training datasets to enable widespread biomechanical applications. Pose estimation algorithms trained on manually labelled datasets have the potential to outperform marker-based methods, through elimination of soft tissue artefact and inter-session variability [41]. However, to achieve this, future manually labelled image datasets need to have three points on each segment [21], employ labellers with anatomical knowledge [3, 42, 43], and include additional detail (e.g. classifying points as occluded or visible). Unfortunately, the cost and time required to manually label datasets is highly prohibitive, as such, alternative methods for creating datasets are being explored. A potential area of research employs marker-based motion capture to obtain anatomically accurate joint and keypoint locations of interest. To date, there have been no critical review of methods to create labelled datasets using marker-based data, which is key to informing best practices moving forwards. Before we explore this area in more detail, it should be noted that any pose estimation method trained on marker-based data will have known marker-based errors [30–32] added to the markerless solution, such as marker placement error [44] and soft-tissue artefact [44, 45]. Thus, while the creation of datasets using marker-based motion capture could be cheaper, faster and more flexible to advances in machine learning such as volumetric modelling [46, 47], we need to carefully consider how close markerless systems should match marker-based results.

One method to create labelled 2D images is by recording marker-based and regular video data simultaneously, and then reprojecting marker-based keypoints into the video camera views. However, there are two major drawbacks of this method. Firstly, the pose estimation algorithm will be trained on images of markered participants, and therefore, the algorithm may generalise poorly to new video images that do not include markers. Secondly, marker-based data is generally collected in highly constrained laboratories and therefore the environments and movements within the new training dataset may be limited. Vafadar, Skalli [28] employed a pose estimation algorithm [48] that was first trained on the marker-based Human 3.6M open access dataset [14], which was then refined using transfer learning on the ENSAM clinical dataset [28]. The ENSAM dataset is of particular interest as it combines 3D marker-based motion capture, markerless video and biplanar x-ray (EOS) to ensure accurate identification of joint centre locations [28, 49]. Vafadar et al. [49] observed significantly improved results after refinement, with considerably decreased systematic differences. However, Human 3.6M [14], ENSAM [28] and the evaluation performed by Vafadar et al. [49] all employed marker-based motion capture videos in highly controlled laboratory environments. Thus, their paper likely had a bias of markerless joint angles towards marker-based results, with additional uncertainty surrounding generalisability of their pose estimation algorithm to video images without markers. Furthermore, they performed a standard Bland-Altman analysis and did not describe how they adjusted their data to deal with repeated measures (S1 Appendix).

A potentially more promising solution is to create realistic synthetic images [47, 50, 51] using virtual environments and body models from which 2D labelled images can be generated [47, 52]. As a case study, Mundt, Oberlack [50] used marker-based data to generate synthetic images of virtual human models performing a sidestepping task in three participants. While no formal analysis was performed, they demonstrated similar markerless results between real and synthetic images, although this was only performed for the lower limbs with a blank grey background. Creation of synthetic images using marker-based data must contend with the previously described marker-based and markerless errors, in addition domain gap errors [47], which are caused by generated images not looking completely real [53], or not accounting for the full range of human movement (e.g. hinge joint at the knee in the model). This is potentially a hindrance of recent work that has generated images from video games [54], although no biomechanical validation has yet been performed on pose estimation algorithms trained on

such datasets. The AGORA training and evaluation dataset has been developed specifically for this purpose, enabling generation of images with diverse movements, environments and cameras angles [47], although it does not yet include biomechanically driven human body models. However, synthetic images using 3D body models have the potential to move beyond sparse pose estimation, as all points on the body are known, potentially facilitating applications of dense pose estimations [55]. Future work is needed to determine the biomechanical accuracy of these synthetic images, especially in the clinical and sporting field where movements are so diverse. As the field of synthetic images is still very novel, it will likely play a significant role in the development of future markerless motion capture training datasets.

## Limitations

While this study has assessed the agreement between markerless motion capture and the 3D marker-based method, inherent marker-based errors such as soft tissue artefact [30, 31], marker placement [41, 44] and altered participant movement patterns [45] will obscure the markerless results. As such determining the true accuracy of markerless systems is difficult due to scarcity and complexity of gold standard methods such as bi-planar radiographic imaging [28, 56] and videoradiography [31, 57]. This should not have affected our comparison between occluded and visible joint centres, however the agreement we observed may differ from that obtained against gold standard methods. It should be noted that the marker-based method included in this study, and all the proposed gold standard methods are performed indoors in laboratory environments, which may differ from outdoor or actual application scenarios. Thus, future work is needed to explore the validation of markerless methods in different environments and with lower than 'research grade' technology. However, a first key step was to ensure that the markerless system can work comparably to marker-based in a laboratory-based setting. While our study had participants wearing tight clothing so that marker-based motion capture could be used, this is generally not possible in clinical environments. However, recent work has demonstrated that the Thiea markerless system is able to handle day-to-day clothing and even baggy pants relatively well [41], which should also hold true for the OpenPose pose estimation algorithm, and thus, day-to-day clothing should not alter our results substantially. As outlined in our S1 Appendix, this study made a comparison of 13231 data points (130 trials x 101 data points) which is more than sufficient to make comparisons between motion capture methods. However, it should be noted that data was collected on relatively young healthy adults, with only 15 participants included in this study, and thus these results may be restricted in their generalisability to examining population groups that differ substantially from our participants (e.g. overweight participants, amputees or children).

## Conclusion

Our results add to the growing body of literature highlighting potential errors associated with open-source pose estimation datasets and algorithms such as OpenPose, demonstrating a need to discriminate between occluded and un-occluded joints. Our findings suggest that for clinical, sporting and research applications during walking, 2D sagittal and frontal plane kinematics identified by the OpenPose algorithm could be obtained for the visible knee and hip joint angles during stance, as they are near or within previously described 3D marker-based motion capture error ranges. Alternatively, the sagittal ankle angle is likely only usable during early and mid-stance, and the frontal plane ankle angle differences were too high across the whole stride and therefore should not currently be used in clinical or sporting applications. Future development to improve the datasets on which markerless pose estimation algorithms are trained could potentially be performed using the wide body of marker-based motion capture

data that has already been collected across the world. By employing marker-based motion capture to drive movement of human models in virtual environments, synthetic images could be generated that provides detailed information of the movement and rotation of all body segments. However, given the known problems within marker-based motion capture, care must be taken before employing them for both training and evaluating markerless motion capture systems.

## Supporting information

**S1 Appendix. This appendix outlines in-depth detail and rationale for the repeated measures Bland-Altman analysis, including equations and examples.** This method was performed to calculate the SD, and subsequently the upper and lower limits of agreement (95% confidence interval), using total variance across all data points from a one-way ANOVA (participant and residual mean square scores).
(DOCX)

**S1 Data. This document includes data used in the S1 Appendix to provide an example for how to perform the repeated measures Bland-Altman analysis.** This example data is the occluded ankle joint angle in the sagittal plane.
(XLSX)

**S2 Data. This appendix encompasses time normalised (101 points) 2D joint centre locations for every trial (130 trials across 15 participants), in both the sagittal and frontal plane, for the markerless system and the reprojected marker-based motion capture system.** The JSON file format is universally implemented.
(JSON)

**S1 Table. This document includes additional Bland-Altman results for joint centre locations in the frontal and sagittal view.**
(DOCX)

**S1 Fig. This document includes additional visual results of joint angle throughout the stride in the frontal and sagittal view.**
(DOCX)

## Acknowledgments

We would like to thank Andrew Chapman from the Maths Resource Centre (MASH) at the University of Bath for his valuable assistance in performing the repeated measures Bland-Altman analysis.

## Author Contributions

**Conceptualization:** Logan Wade, Laurie Needham, Murray Evans, Polly McGuigan, Steffi Colyer, Darren Cosker, James Bilzon.

**Data curation:** Logan Wade, Laurie Needham, Murray Evans.

**Formal analysis:** Logan Wade.

**Funding acquisition:** Darren Cosker, James Bilzon.

**Investigation:** Logan Wade, Laurie Needham.

**Methodology:** Logan Wade, Laurie Needham, Murray Evans.

**Project administration:** Logan Wade, Polly McGuigan.

**Software:** Murray Evans.

**Visualization:** Logan Wade.

**Writing – original draft:** Logan Wade.

**Writing – review & editing:** Logan Wade, Laurie Needham, Murray Evans, Polly McGuigan, Steffi Colyer, Darren Cosker, James Bilzon.

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
