## [Decision Letter · Decision Letter 0]

12 Mar 2023

PONE-D-23-04720Examination of 2D frontal and sagittal markerless motion capture: Implications for 2D and 3D markerless applicationsPLOS ONE

Dear Dr. Wade,

Thank you for submitting your manuscript to PLOS ONE. After careful consideration, we feel that it has merit but does not fully meet PLOS ONE’s publication criteria as it currently stands. Therefore, we invite you to submit a revised version of the manuscript that addresses the points raised during the review process.

We look forward to receiving your revised manuscript.

Kind regards,

Yaodong Gu

Academic Editor

PLOS ONE

Journal Requirements:

Additional Editor Comments:

The limitation part shall be added.

Reviewers' comments:

Reviewer's Responses to Questions

**Comments to the Author**

1. Is the manuscript technically sound, and do the data support the conclusions?

Reviewer #1: Yes

Reviewer #2: Yes

2. Has the statistical analysis been performed appropriately and rigorously? 

Reviewer #1: Yes

Reviewer #2: I Don't Know

3. Have the authors made all data underlying the findings in their manuscript fully available?

Reviewer #1: Yes

Reviewer #2: Yes

4. Is the manuscript presented in an intelligible fashion and written in standard English?

Reviewer #1: Yes

Reviewer #2: Yes

5. Review Comments to the Author

Reviewer #1: The purpose of this study was to examine if occluded joint locations from markerless motion capture produced 2D joint angles with reduced accuracy compared to visible joints.

From a clinical point of view, the aim of the study is useful. It is necessary to have precise low-cost devices that allow movement to be analysed.

The authors use the OpenPose algorithm. The motion model is the Kalman filter. The marked-based data uses Qualisys and Visual3D.

The methodology is well described, I thank the authors for the detailed explanations.

In the results, the authors should describe that the limits of agreement according to the Bland and Altman plots are wide, which shows a significant error or bias for the detection of ankle angle, but also exists for the hip joint.

The discussion is correct and does a detailed analysis of the results, showing the limitations of OpenPose and describing alternatives. I recommend that the authors make recommendations for clinical staff.

Reviewer #2: This paper compares sagittal joint angles for occluded vs visible joints for a single camera 2D markerless motion capture system as well as frontal plane joint angles to marker-based motion capture in 15 participants. The paper includes a lengthy discussion of issues with marker-based motion capture and theoretical approaches to developing training datasets for 2D solutions.

There are some issues with the use of correct terminology for accuracy/precision/reliability.

Some methodological details are missing and the paper lacks a discussion of limitations of their own work.

Keywords: consider adding “markerless motion capture” if possible

Title – Would recommend “Examination of 2D frontal and sagittal markerless motion capture” as the title.

Abstract

Line 23 - Please be more specific about the markerless motion capture system used (i.e. 2D markerless motion capture from a single camera view).

Line 35 – are these only marker-based methods?

Introduction

Starting the paper with issues with marker-based motion capture seems sub-optimal for a paper that has a primary aim of comparing markerless against a marker-based reference. This paper may be better motivated without the initial discussion of marker-based problems, which would be more appropriate in a limitations section. Additionally, the authors could note that while marker-based motion capture may not necessarily accurately reflect the motion of the underlying skeletal anatomy, as the aim of the first objective was to compare visible vs occluded 2D markerless motion capture, 3D marker-based is expected to perform equally well on both sides, enabling the markerless occluded vs visible comparison.

Furthermore, the discussion of the lack of access to x-ray-based methods to calculate markerless accuracy may not be helpful. These approaches still have errors. This discussion implies that markerless systems that do not have equivalent “accuracy” to x-ray-based methods are not useful, which I’m guessing is not the intent.

Line 39 – Please consider if “accuracy and reliability” are the correct terms here and throughout the paper (see further comments on accuracy for Lines 77-78, 80)

Line 50 - Please provide further motivation for the prospective use of single camera markerless motion capture beyond “without the need for specialist personnel to process the data”

Lines 52-56 – The wording here implies that the 2 studies mentioned were using an algorithm that identifies occluded joint locations, but that does not appear to be the case. It is not entirely clear why these 2 studies are mentioned (are these the only studies that have reported sagittal plane kinematics using a single camera 2D markerless system?) or why you discuss different algorithms than those used by these 2 studies.

Line 61 – please remove “inevitable”

Lines 68 – 69 – “low random differences” from what? Again, consider the correct terms to use (if precision and reliability are intended to mean the same thing, standardize your terminology)

Line 77 – 78 – Reconsider the use of the term “accurate” in your objectives since you argue that your reference technology (marker-based) is not accurate and therefore you lack a true value against which to measure.

Line 80 – Define here or in the methods your definitions of “too variable” and “meaningful changes” against which your results are compared or otherwise define the threshold for acceptance of “accurate” markerless kinematic results. Note that “too variable” indicates a lack of precision, not a lack of accuracy.

The further objective of discussing methods of creating improved markerless training data is missing.

Methods

Please define acronyms (i.e. MTP, TTL) at the first instance

Line 86 – How was constant speed achieved? Was it the same for all participants?

Lines 111 – 133 - The details of the tracking method to identify the participant across multiple frames would be appropriate for a supplement.

Line 142 – 145 - Visual3D does not provide recommendations for joint location approaches, but several options. Please indicate the approaches used and reference the original papers for the chosen approach.

Line 152 – Replace “high-quality” with “aligned” or similar

Results

Specify the number of trials and strides recorded and how many were included in analysis.

Were any data lost due to problems with marker-based or markerless tracking?

Please report the walking speed.

Differences in pixels are not very meaningful as they depend on the camera resolution and the distance of the participants to the cameras, neither of which are provided.

Line 201 – The term “absolute random differences” is not meaningful. Why not compare the limits of agreement?

Discussion

Line 272 – “the relative proportion of error to signal is lower.” Should this be: “higher” instead of lower? (higher proportion of error relative to signal)

Line 290 – unclear how the conclusion of “discounting occluded joint results” is reached when there has been no definition of acceptability.

Line 311 – I would encourage the authors to consider whether equivalent accuracy to biplanar x-ray is the goal of markerless or if the data may still be useful if it doesn’t not achieve the same resolution. Further, having a reference methodology that is less prone to issues such as soft tissue movement does not eliminate differences due to differing coordinate system definitions.

Line 318 – Be specific in your discussion of implications for 3D motion capture systems. Are these statements true for all 3D markerless systems or only the ones you are familiar with?

Line 336 – 354 – Unclear of the relevance of this discussion to the current paper.

Line 351 – 353 – Would caution against predicting the results of another study without analysis of the data.

Line 425: Unclear what is meant by “loosen data setup restrictions common to 2D motion capture.”

There are no limitations of your study discussed. Limitations of marker-based motion capture would be appropriate in such a section. Please include discussion of limitation of clothing. In clinical applications, it is unlikely the participants will be wearing minimal spandex. Please also discuss limitations related to the B&A approach (From Bland JM & Altman DG. 2007: “It is an implicit assumption that the difference between the two methods is reasonably stable across the range of measurements” – Is that true for this study?)

Please include limitations of sample size and a young healthy sample

Unlikely to have force data in clinical/sporting applications. How would the use of kinematic gait events affect the data?

Figure 4 A – plot is truncated

Table 1 – define LOA

Supplement 1 – should this be referring to Supplement 2 (not 3) for tables, data, etc?

Supplement 2 - unclear what the sample data represents (i.e. are these joint angles? Which joint?)

Supplement 3 Figure 1 – please label vertical axis as “Differences in Angle”

E – unclear why the L/R sides are compared

6. PLOS authors have the option to publish the peer review history of their article (what does this mean?). If published, this will include your full peer review and any attached files.

Reviewer #1: No

Reviewer #2: No

---

## [Author Response · Author response to Decision Letter 0]

21 Mar 2023

We would like to thank the editor and reviewers for their time and effort in improving this manuscript. We have included a document which responds to each comment made by the Editor and both Reviewers

---

## [Decision Letter · Decision Letter 1]

22 May 2023

PONE-D-23-04720R1Examination of 2D frontal and sagittal markerless motion capture: Implications for 2D and 3D markerless applicationsPLOS ONE

Dear Dr. Wade,

Thank you for submitting your manuscript to PLOS ONE. After careful consideration, we feel that it has merit but does not fully meet PLOS ONE’s publication criteria as it currently stands. Therefore, we invite you to submit a revised version of the manuscript that addresses the points raised during the review process.

We look forward to receiving your revised manuscript.

Kind regards,

Yaodong Gu

Academic Editor

PLOS ONE

Additional Editor Comments:

The methods part shall be more detailing to reflect how you carry out the testing.

Reviewers' comments:

Reviewer's Responses to Questions

**Comments to the Author**

1. If the authors have adequately addressed your comments raised in a previous round of review and you feel that this manuscript is now acceptable for publication, you may indicate that here to bypass the “Comments to the Author” section, enter your conflict of interest statement in the “Confidential to Editor” section, and submit your "Accept" recommendation.

Reviewer #2: (No Response)

Reviewer #3: (No Response)

2. Is the manuscript technically sound, and do the data support the conclusions?

Reviewer #2: Partly

Reviewer #3: Yes

3. Has the statistical analysis been performed appropriately and rigorously? 

Reviewer #2: No

Reviewer #3: Yes

4. Have the authors made all data underlying the findings in their manuscript fully available?

Reviewer #2: Yes

Reviewer #3: No

5. Is the manuscript presented in an intelligible fashion and written in standard English?

Reviewer #2: Yes

Reviewer #3: Yes

6. Review Comments to the Author

Reviewer #2: The authors still seem to be missing correct definitions of terms critical to their study, specifically accuracy.

In measurement terms, accuracy refers to how close measured values are to “true” values. As true values are difficult to determine in many cases, the Bland & Altman approach was developed specifically to compare 2 measurement systems (to determine their interchangeability, i.e. limits of agreement) in the absence of true values. As such, the Bland & Altman method cannot be used to assess accuracy. The authors can choose to (1) define marker-based motion capture results as the true values and then calculate the accuracy of the markerless system OR (2) they can use the Bland & Altman approach to compare the offset and limits of agreement between markerless and marker-based motion systems, but in this case they cannot report accuracy.

I will repeat my comments from my previous review, specifically:

"Lines 75 – 80 - Reconsider the use of the term “accurate” in your objectives since you argue that your reference technology (marker-based) is not accurate and therefore you lack a true value against which to measure.

Line 80 - Note that “too variable” indicates a lack of precision, not a lack of accuracy."

They should also stick with the terms defined by Bland & Altman (limits of agreement) instead of using “random difference” which will not be meaningful to most readers.

Literature discussed in the introduction remains subpar. The inclusion of articles that did not measure frontal joint angles is confusing.

Basic methodological pieces are missing, i.e. recording frequencies, image resolutions, if joint locations defined in the same way for both systems.

The methods of how front plane joint positions are measured in both systems are not described.

The authors have not justified reporting absolute differences in pixels, which seems like it will not be meaningful, as the size of the joint will change in every frame as the participant moves further from or closer to the camera.

Calculations of joint angles and percentages are missing from the methods.

Figure 2 sub plots are not identified

Figure captions are incomplete (i.e. lines and shaded regions indicate what?)

I maintain that the title is an overreach. It is unclear how your conclusions apply to 3D analysis.

This paper requires significantly more work and would benefit from careful reviewing and editing from the authors. I am concerned that after a first round of reviews, accuracy is still not correctly used. Additionally, there are still missing methodology components and incomplete figure captions.

Reviewer #3: The study offers important insights into the strengths and limitations of markerless motion capture, and specifically sheds light on its limitations in accurately capturing occluded joint angles, particularly in the sagittal plane. In addition, here are some other potential issues that should be considered for the next revision:

1. In the introduction section, it would be beneficial to provide a brief overview of the background and current status of 2D and 3D marker-based motion capture techniques, and explain why these techniques are crucial in the analysis of human motion. Additionally, it is suggested to further clarify the research purpose and research questions to help readers better understand the motivation and value of this paper.

2. It would be easier for readers to understand if you could explain the difference between occluded joints and visible joints.

3. It is recommended to describe the specific algorithms or techniques used for markerless 2D and 3D motion capture, ensuring sufficient details are provided to allow readers to understand your experimental setup and data collection process.

4. In the discussion part, it is recommended to provide a brief description of the aim and main findings in the first paragraph of the manuscript.

5. In the conclusion part, please also show relevant descriptions about the contributions for future clinical or scientific research.

6. Furthermore, the future research direction and prospects are also an important part of the article, and it would be beneficial to expand on this section and provide more detailed content.

7. PLOS authors have the option to publish the peer review history of their article (what does this mean?). If published, this will include your full peer review and any attached files.

Reviewer #2: No

Reviewer #3: No

---

## [Author Response · Author response to Decision Letter 1]

16 Jun 2023

We have attached a word document which addresses all reviewers comments. All line comments in the reply to reviewers docuement relate to the tracked changes document.

---

## [Decision Letter · Decision Letter 2]

31 Jul 2023

PONE-D-23-04720R2Examination of 2D frontal and sagittal markerless motion capture: Implications for markerless applicationsPLOS ONE

Dear Dr. Wade,

Thank you for submitting your manuscript to PLOS ONE. After careful consideration, we feel that it has merit but does not fully meet PLOS ONE’s publication criteria as it currently stands. Therefore, we invite you to submit a revised version of the manuscript that addresses the points raised during the review process.

We look forward to receiving your revised manuscript.

Kind regards,

Yaodong Gu

Academic Editor

PLOS ONE

Journal Requirements:

Additional Editor Comments:

Please check the kinematic data questions raised by the reviewer.

Reviewers' comments:

Reviewer's Responses to Questions

**Comments to the Author**

1. If the authors have adequately addressed your comments raised in a previous round of review and you feel that this manuscript is now acceptable for publication, you may indicate that here to bypass the “Comments to the Author” section, enter your conflict of interest statement in the “Confidential to Editor” section, and submit your "Accept" recommendation.

Reviewer #3: All comments have been addressed

Reviewer #4: All comments have been addressed

2. Is the manuscript technically sound, and do the data support the conclusions?

Reviewer #3: Yes

Reviewer #4: Yes

3. Has the statistical analysis been performed appropriately and rigorously? 

Reviewer #3: Yes

Reviewer #4: Yes

4. Have the authors made all data underlying the findings in their manuscript fully available?

Reviewer #3: Yes

Reviewer #4: Yes

5. Is the manuscript presented in an intelligible fashion and written in standard English?

Reviewer #3: Yes

Reviewer #4: Yes

6. Review Comments to the Author

Reviewer #3: (No Response)

Reviewer #4: In this work, the authors investigated the accuracy of 2D joint angles obtained from occluded joint locations in markerless motion capture compared to visible joints. The author recruited 15 participants to recorded while walking using marker-based and machine vision cameras. The results showed that markerless joint angles on the occluded side had higher variation and disagreement compared to visible joints. However, the visible joint angles in the sagittal plane were relatively accurate and consistent with marker-based measurements. On the frontal plane, the markerless joint angles had acceptable agreement with marker-based data for the hip and knee but not for the ankle. The study also identified limitations in current pose estimation algorithms and datasets, suggesting the need for improved methods to create anatomically accurate markerless training data using marker-based motion capture. This work could be accepted after minor revisions. Other questions were shown below.

1.The study only included 15 healthy participants, which is a relatively small sample size. A larger sample size can improve the reliability and generalization ability of the results.

2. This study used indoor marker cameras and machine vision cameras for data collection, which may differ from outdoor or actual application scenarios. Therefore, more exploration and validation are needed for applications in different environments.

3. This study mainly uses the Bland Altman analysis to evaluate the consistency, but other metrics are also worth considering, such as correlation coefficient or Root-mean-square deviation. Combining multiple evaluation indicators can provide a more comprehensive analysis of results. Have other statistical analyses been conducted to evaluate the reliability and significance of the results, in addition to the Bland Altman analysis?

4. Can we further explain or discuss why the unmarked deviation standard deviation and consistency limit of the occluded joint on the Sagittal plane are twice that of the visible joint? May this difference be related to the characteristics of pose estimation algorithms?

5. In the discussion, can we explore the shortcomings of current pose estimation algorithms and labeled datasets, and propose suggestions for improvement or future research directions?

6. Considering that the consistency limit of ankle joint is still higher than the requirements of practical application, can you provide suggestions or ideas on how to optimize unmarked Motion capture of ankle joint？

7. PLOS authors have the option to publish the peer review history of their article (what does this mean?). If published, this will include your full peer review and any attached files.

Reviewer #3: No

Reviewer #4: No

---

## [Author Response · Author response to Decision Letter 2]

23 Aug 2023

Reviewer 3 Comments

No Comments

Reviewer 4

In this work, the authors investigated the accuracy of 2D joint angles obtained from occluded joint locations in markerless motion capture compared to visible joints. The author recruited 15 participants to recorded while walking using marker-based and machine vision cameras. The results showed that markerless joint angles on the occluded side had higher variation and disagreement compared to visible joints. However, the visible joint angles in the sagittal plane were relatively accurate and consistent with marker-based measurements. On the frontal plane, the markerless joint angles had acceptable agreement with marker-based data for the hip and knee but not for the ankle. The study also identified limitations in current pose estimation algorithms and datasets, suggesting the need for improved methods to create anatomically accurate markerless training data using marker-based motion capture. This work could be accepted after minor revisions. Other questions were shown below.

REPLY: We thank the reviewer for taking the time to read our manuscript. We believe that we have addressed all the reviewer comments and the clarity of the manuscript is now further improved. Please note that all line changes referred to below are from the tracked changes document.

The study only included 15 healthy participants, which is a relatively small sample size. A larger sample size can improve the reliability and generalization ability of the results

REPLY: We agree with the reviewer that the findings from 15 healthy participants may not be widely applicable to other populations, e.g., clinical populations, and certainly more work is needed in the future to ensure that the same agreement can be achieved in those populations. This was touched on in the limitations section, however this section has been expanded to further describe the limitations of only including 15 healthy participants. Line 452-455. Nonetheless, we do feel that the estimates of agreement presented in this paper based on this sample do provide a very useful addition to the literature upon which future research can build.

This study used indoor marker cameras and machine vision cameras for data collection, which may differ from outdoor or actual application scenarios. Therefore, more exploration and validation are needed for applications in different environments.

REPLY: We fully agree that further exploration and validation is needed in less controlled scenarios and with lower than ‘research grade’ technology. However, a first key step was to ensure that the markerless system can work comparably to marker-based in a laboratory-based setting. We have added this to the limitations section. Line 440-446

This study mainly uses the Bland Altman analysis to evaluate the consistency, but other metrics are also worth considering, such as correlation coefficient or Root-mean-square deviation. Combining multiple evaluation indicators can provide a more comprehensive analysis of results. Have other statistical analyses been conducted to evaluate the reliability and significance of the results, in addition to the Bland Altman analysis?

REPLY: We used a repeat measures Bland-Altman analysis, as the standard bland-Altman analysis assumes all data points are independent, which is not true for our dataset that has repeated measures. This is a similar assumption for standard correlation coefficient and root-mean-square deviation analysis. As such to use these methods, the data generally needs to be averaged multiple times to meet this assumption, drastically reducing the variability. However, we do agree that other metrics are worth considering, thus we have included a repeated measures correlation statistical analysis to Table 1, which provides a correlation coefficient value between marker-based and markerless motion capture for repeated measures joint angles. This analysis is not possible to calculate for pixel locations, given that the normalised resultant pixel differences we report are made up of a 2D pixel locations (x and y coordinates). It would not make sense to break down the resultant pixel locations into their 2D values for these repeated measures correlation analysis alone. Methods for the repeat measures correlation coefficient have been added to the methods (Line 208-210), and implications of these results has been added throughout the results and discussion sections.

Can we further explain or discuss why the unmarked deviation standard deviation and consistency limit of the occluded joint on the Sagittal plane are twice that of the visible joint? May this difference be related to the characteristics of pose estimation algorithms?

REPLY: This has already been discussed at length on Lines 62-65 356-360, 376-381

In the discussion, can we explore the shortcomings of current pose estimation algorithms and labeled datasets, and propose suggestions for improvement or future research directions?

REPLY: This is a crucial aspect of the paper, which was already explored in depth on Lines 356-431 

Considering that the consistency limit of ankle joint is still higher than the requirements of practical application, can you provide suggestions or ideas on how to optimize unmarked Motion capture of ankle joint？

REPLY: Without improving the labelling in markerless datasets, this will be a challenging task. We have outlined methods for potentially improving these datasets in Lines 375-431. We have also added a sentence to the discussion on lines 320-323 which outlines a potential short-term solution that fuses markerless motion capture with IMU devices to improve accuracy at the ankle.

---

## [Decision Letter · Decision Letter 3]

23 Oct 2023

Examination of 2D frontal and sagittal markerless motion capture: Implications for markerless applications

PONE-D-23-04720R3

Dear Dr. Wade,

We’re pleased to inform you that your manuscript has been judged scientifically suitable for publication and will be formally accepted for publication once it meets all outstanding technical requirements.

Kind regards,

Yaodong Gu

Academic Editor

PLOS ONE

Additional Editor Comments (optional):

The final response has arrived, and I agree with the results. Now it could be accepted.

Reviewers' comments:

Reviewer's Responses to Questions

**Comments to the Author**

1. If the authors have adequately addressed your comments raised in a previous round of review and you feel that this manuscript is now acceptable for publication, you may indicate that here to bypass the “Comments to the Author” section, enter your conflict of interest statement in the “Confidential to Editor” section, and submit your "Accept" recommendation.

Reviewer #5: All comments have been addressed

2. Is the manuscript technically sound, and do the data support the conclusions?

Reviewer #5: Yes

3. Has the statistical analysis been performed appropriately and rigorously? 

Reviewer #5: Yes

4. Have the authors made all data underlying the findings in their manuscript fully available?

Reviewer #5: Yes

5. Is the manuscript presented in an intelligible fashion and written in standard English?

Reviewer #5: Yes

6. Review Comments to the Author

Reviewer #5: I would like to thank the authors work on the revision of this manuscript. This manuscript has been well revised and could be published.

7. PLOS authors have the option to publish the peer review history of their article (what does this mean?). If published, this will include your full peer review and any attached files.

Reviewer #5: No

---

## [Editor Report · Acceptance letter]

2 Nov 2023

PONE-D-23-04720R3 

Examination of 2D frontal and sagittal markerless motion capture: Implications for markerless applications 

Dear Dr. Wade:

I'm pleased to inform you that your manuscript has been deemed suitable for publication in PLOS ONE. Congratulations! Your manuscript is now with our production department. 

Kind regards, 

on behalf of

Professor Yaodong Gu 

Academic Editor

PLOS ONE